# When COVID-19 Is the Invader and Internal Communication Is the Hero: Understanding the Influence of Internal Communication on Individual Performance and Evaluating the Mediating Role of Perceived Support

**Daniel Roque Gomes** [1,2,*] , **Patricia Lourenço** [2] **and Neuza Ribeiro** [3]

1 ICNOVA—NOVA Institute of Communication, 3030-329 Coimbra, Portugal
2 School of Education (ESEC-IPC), Polytechnic Institute of Coimbra, R. Dom João III, 3030-329 Coimbra, Portugal; patricialourenco64@gmail.com
3 CARME—Centre of Applied Research in Management and Economics, School of Technology and Management, Polytechnic Institute of Leiria, Campus 2—Morro do Lena, Alto do Vieiro, 2411-901 Leiria, Portugal; neuza.ribeiro@ipleiria.pt
* Correspondence: drmgomes@esec.pt

**Abstract:** Objective: The main objective of this study was to evaluate the impact of internal communication (IC) of organisations on the quality of the individual–organisation relationship, specifically with regard to the effects generated with respect to individual performance (IP) in a pandemic framework. In this sense, the study intends to evaluate the effects of IC on the employee's IP, having as a mediator perceived organisational support (POS). Methodology: To achieve the aforementioned objectives, a cross-sectional quantitative study was prepared, data for which were collected during a period of confinement that took place between 9 February and 15 March 2021. A total of 340 individuals of both sexes participated in the study. Genders were 67.6% female and 32.4% male, with ages ranging from 25 years to over 61 years, from all districts of Portugal and the Islands. Results: The main results obtained showed that IC was positively and significantly correlated with POS and also with IP, and that there was also a total mediation effect of POS in the relationship between IC and IP. Practical implications: These results seem to support the need for organisations to invest in their internal communication practices as a way of stimulating strong and fruitful relationships between workers and the organisation. Internal communication seems to be a relevant indicator for the management of proximity relationships with workers, especially in adverse contexts, like the ones experienced during the pandemic crisis. Well-developed internal communication supports and practices seem to be a valid path towards developing bonds leading to improved performance.

**Keywords:** internal communication; perceived support; performance

## 1. Introduction

In December 2019, hospitals in the Chinese city of Wuhan recorded the first cases of pneumonia with an unknown cause, which was later confirmed to be an acute respiratory disease caused by the Sars-Cov-2 virus, a new type of coronavirus (Mesquita et al. 2020). On 30 January 2020, with a growing number of infections inside and outside China, the World Health Organisation (WHO) declared a global public health emergency (Velavan and Meyer 2020). On an economic level, COVID-19 caused the worst global recession since 1930 (Shen et al. 2020). Within the context of an economic crisis, organisations have been subject to several challenges, including the strategic planning they will use to deal with their employees, taking into account the impact of COVID-19 on the work context. As a result, aspects such as organisational coordination, management processes or productivity have become fundamental. Communication has become digital and telecommuting, regardless of the family context, has been and continues to be a new reality for many employees

(DeFilippis et al. 2020). It is in this context that other issues, namely in relation to how the organisation establishes perceived organisational support (POS), maintaining a positive record of well-being, and individual performance (IP) within the new context. To find answers to these uncertainties, it is important to analyse the relevance of the internal communication (IC) approach to the individual–organisation relationship.

Despite the diversity of perspectives in the scientific literature on the phenomenon of IC, within an organisation, it can be considered to function as a set of communicative processes employed with the intent of transmitting information to its public and creating, developing and sustaining group, shared and symbolic awareness (Neto and Cruz 2017). Thus, IC is a relevant area of activity that has acquired strategic importance in organisations, since it is able to involve all stakeholders by means of cohesive communication, leading to the involvement of employees, the strengthening of ties, and encouragement of the building and sharing of knowledge among them (Alves and Souza 2015). It is in this sense that one can question the extent to which organisations remain concerned about the stability, emotional comfort and support provided to their employees in a pandemic context characterised by reactive and operational decisions in daily business management. It also seems relevant to consider the role that IC can play in this functional or emotional monitoring framework between the employee and the organisation. IC may be directly related to effective support, resulting in an area that has a natural place, albeit one insufficiently explored, in verifying the type of relationships and associations that are considered here. This study's main research question relates to understanding the predictive effect of IC on IP, and whether POS has a mediating effect on this relationship within the pandemic context brought about by COVID-19. Furthermore, this study highlighting the discussion of IC against the backdrop of COVID-19 becomes relevant because it is a subject of limited scientific study with a lack of significant conclusions regarding the relationship between organisations and collaborators.

The analysis of these two points presented a clear invitation to compile a study investigating the role of IC in IP and POS. Within this context, the present study proposes to analyse the effects and strategy of IC with the organisation–collaborator relationship.

## 2. Literature Review

*Internal Communication, Organisational Support and Individual Performance*

The outbreak of COVID-19 caused the closure of schools and workplaces in order to mitigate the severity of the spread of the disease, leading organisations to a forced adaptation to this new reality (Hamouche 2020). In addition, it has caused hundreds of thousands of deaths, tested the limit of health systems, and generated a large number of unanswered questions globally. Governments, societies and organisations are enduring a crisis and are looking for ways around the destruction that COVID-19 has caused them (Dirani et al. 2020). One of the measures implemented by organisations in order to allow businesses to move forward was the adoption of remote working; despite being a practice already used by many, with the pandemic, its use was almost considered mandatory (Losekann and Mourão 2020). This impact has resulted in several severe impacts on individuals' personal relationships (Venkatesh 2020), meaning that in addition to organisations' concerns regarding the recovery of their business and better managing the pandemic, communication with employees (internal communication) has also become a priority in terms of providing them with continuous support (Dirani et al. 2020).

In the context of this scenario, many organisations face various communication problems due to the changing environment caused by COVID-19 (Dirani et al. 2020). Thus, it makes sense to talk about solutions that can perhaps address some of these problems; therefore, we present the idea of Grunig (1984, 1989), who proposed the concept of asymmetrical and symmetrical communication. Grunig considered that asymmetrical communication to be unbalanced, since only the information coming from the top down is relevant (Murphy 1991). Over the years this idea has matured through theoretical and empirical development, with authors eventually instituting a symmetrical communication model, which emerges

as a reflection of the interests of the organisation and the collaborator, where both sides are respected in a symmetrical manner (Men and Stacks 2014). Thus, symmetrical internal communication involves the notions of openness and reciprocity, negotiation, and tolerance in disagreements between the organisation and the employee. The symmetrical model can thus be implemented by organisations in order to solve problems and achieve mutual agreement. The goal is to promote ideas and behaviours that can be respected and accepted by all involved (Yue et al. 2020).

From this perspective, in the context of the diversity of factors that explain the effectiveness and efficiency of organisations in various moments of their activity (e.g., Gomes and Neves 2010; Gomes and Neves 2019; Chatzopoulou et al. 2021), IC is an important foundation of modern organisations (Men and Stacks 2014), and is regarded, on the one hand, as an instrument for developing the commitment of employees to the organisation, and, on the other, as an instrument for the strategic construction of organisational management (Neto and Cruz 2017). According to Men (2014), IC functions as a central process in which employees share information, and create bonds and relationships, thus helping to build the culture and values of the organisation. This happens when employees identify themselves with the organisation to the extent that they recognise it as their own, eventually internalising and assuming the culture of the organisation, its values, beliefs and objectives (Yue et al. 2020). For Brandão (2018), IC is a system of interaction that allows the sharing among employees of meanings that reflect the concept of the company, serving as a reference for individuals (and their behaviour), since it allows them to assimilate and reinforce the values of the organisation.

Men and Stacks (2014) report that IC demonstrates effectiveness in developing employee attitudes, namely trust, commitment, organisational identification, job satisfaction, and positive relationships between colleagues, leading to greater productivity, individual performance, learning, communication and improvement of external relations. The same authors also note the importance of IC in facilitating interaction between organisations and employees, since it helps to create social bonds that are based on meaning and value. From another perspective, Ćorić et al. (2020) regarded IC to include all the communication that occurs within various types of organisations, representing the sharing of ideas, information, attitudes and emotions shared among people, often with the intention of modifying behaviours in order to meet organisational objectives. The influence of IC in organisations is considered to be essential for the success and strengthening of the relationship between the organisation itself and its employees and, according to Brandão (2018), this communication should be strategic and decisive, always focusing on the involvement of individuals in organisations.

Within the context of this representative framework of IC, COVID-19 has given rise, from an empirical perspective, to several questions regarding its capacity to act and how organisations are able to establish relationships of connection and support with their employees in view of the many harmful consequences caused by the pandemic.

IC in organisations should be considered strategically, as a valuable cultural reference for employees, since it can be a decisive and integral element of the organisational system, facilitating the integration of its employees, and acting as an agent of change and internal cohesion (Brandão 2018). Thus, the IC strategy makes a wide range of contributions to the organisation, mainly related to the activities of production and coordination, socialisation and integration, image management and organisational change, and innovation (Gomes et al. 2014). The same authors added that the existence of strategic success at the IC level presupposes that some rules that give rise to benefits and practical effects are applied. An IC strategy should be developed with the objective of sustaining communication guidelines through individual and collective dimensions, taking into account the functional part (tasks) and the contributions of expectations (role). The authors state, through the description of the Henriet and Boneu model, that the strategy should be developed through the axes of IC: (1) understanding; (2) circulation; (3) confrontation; and (4) cohesion. The articulation of these four axes results in the functions and purposes of IC becoming more visible within

an organisation, with the aim of balancing the IC strategy. In order for this balance around the IC axis matrix to occur, organisations must align, adapt and activate the axes, so that the strategy can be planned in a dedicated way and directed toward the objectives and functions foreseen in the organisational context (Gomes et al. 2014).

IC is an internal organisational process that allows the sharing of information, enabling the existence of relationships that involves trust and companionship among employees within the organisation, as well as between managers and the organisation itself. According to Karanges (2014), employees tend to develop several professional relationships within the workplace, and presented two of the main relationships that dominate professional life, as follows: (1) the relationship with the organisation (the entire management team); and (2) the relationship with the direct manager. Both of these relationships are considered to be relationships of social exchange.

The organisation–employee relationship is considered to be part of the internal socialisation process, wherein factors such as communication, performance, satisfaction, well-being, success and support are favourable elements for the strengthening of this connection. For this to happen, one key paradigm in the analysis of relationships in the workplace described in the literature is Social Exchange Theory (SET) (Chernyak-Hai and Rabenu 2018). SET was first presented by Blau (1964), and can be used to illustrate the relationships among humans with respect to attitudes and behaviours (Barbalet 2017). According to Andrade et al. (2017), the assumption of SET is based on the fact that the interactions among individuals or communities can be characterised as an attempt to maximise rewards and reduce costs (material or non-material). The authors also add that, according to Blau (1964) analysis, interactions are maintained because individuals consider them to be more compensatory, regardless of the reason. These interactions are considered from a relationship perspective, where social resources are exchanged through the development of cognition, emotion and behaviour, in order to achieve mutual benefits for individuals, employees, customers, organisations and societies (Karanges 2014; Andrade et al. 2017). Furthermore, SET is also considered a kind of exchange of mutual satisfaction between two parties, which is exchanged through a process of reciprocity, providing psychological benefits that serve to maintain a stable social system (Swalhi et al. 2017). According to Yin (2018), an individual will first carry out an assessment of a possible reward generated by the interaction with other individuals. If neither party ultimately would obtain a satisfactory reward, social exchange does not take place. Therefore, the SET process begins when an individual or organisation, usually a supervisor or a co-worker, relate to each other in a positive or negative way (Cropanzano et al. 2017). These authors assume this to be the first step of SET, which, if it is a positive action, may include organisational support activity or justice. However, if it is a negative action, this may reflect abusive supervision or lack of respect. In addition, there is empirical evidence linking SET to POS, since SET makes it possible to build a sense of support, identification, satisfaction and trust in the employee (Cropanzano et al. 2017).

SET is seen as an exchange of interactions conceived through a central process with relevance to social life, and describes the relationships among individuals, groups and organisations (Andrade et al. 2017). In this sense, favourable social exchange relationships are essential to achieving individual goals and objectives, as all individuals or groups are interdependent on others (Karanges 2014). Based on this idea, and for the purposes of this study, it is important to understand how SET is dealt with from an organisational perspective with respect to support and IC. According to Andrade et al. (2017), SET is one of the most enduring conceptual frameworks, and is widely used for the study of organisational behaviour, with evidence related to commitment, support and supervision having been successfully explored. It is thus considered that when an organisation promises its employees rewards, respect, and justice, a high level of trust will be generated, which will increase the willingness of employees to work more and better, which will be reflected in their IP in relation to the organisation (Yin 2018; Ribeiro et al. 2018). Therefore, the commitment and engagement of employees at work is a function of the rewards they

expect from their employer (Yin 2018), and the link in this connection may be IC, which is strongly related to POS, IP, and productivity (Neto et al. 2018).

According to Smidts et al. (2001), IC facilitates interactions between organisations and employees that create social relationships based on meaning and value. This can increase productivity and drive positive attitudes among employees, based on the resources of social exchange, where individuals use cognitive filters to convey information and actions (positive or negative) (Karanges 2014). As a way of connecting IC and SET, Kurtessis et al. (2017) suggested that POS was strongly dependent on the intentions of employees with respect to the organisation. The authors also recognised POS as a process of social exchange, whereby employees feel obliged to help the organisation achieve goals and objectives.

As a result of the positive effects of support on employees, POS and IC are beginning to be regarded as part of the day-to-day relationship between organisations and employees. As previously noted, COVID-19 has raised several concerns, both in terms of strategic management and the psychological well-being of the employee. In view of this, it is relevant for this study to approach support as a way of addressing the consequences of the pandemic on the employee, given the functioning of SET.

Along the same lines, for Harris and Kacmar (2018), the primary theoretical basis of POS is SET. According to Chen et al. (2020), organisations need social interactions and POS, since employees need organisational support in order to gain motivation to work and, consequently, improve their IP. POS results in positive effects on employees, as it significantly contributes to their satisfaction, organisational commitment and IP: "Whenever organizations provide support to their employees, they perceive their contribution as meaningful, and they contribute more to the organization's productivity" (Rubel et al. 2020, p. 9). It is through POS that an organisation can observe and analyse the various contributions that employees can make to the organisation's success, and the well-being and pleasure of working in that place are aspects that meet an individual's needs to belong, to be respected, and to be approved of by the organisation that employs them (Dursun 2015). Thus, when employees perceive that their organisation provides them with a good basis of support, a feeling of security and confidence towards it is created. In particular, the organisational climate and citizenship behaviour are predictably improved, contributing strongly to organisational success (Park et al. 2020), and also making it possible to assess the degree to which the organisation supports and values its employees in the work context (Yücel et al. 2020).

Regarding consequences related to POS, Karanges (2014) raised the notion of organisational identification, as it has an affective cognitive function that also influences an employee's sense of pride and belonging in the organisation. Neto et al. (2018) also added that, addition to POS, IC and organisational identification are mechanisms that affect the organisation, in that they influence employee well-being, satisfaction and IP (Xiao 2020). However, although the existence of relationships among the various factors addressed in this study has been described, it is important to note that, due to the present reality of the pandemic in which we live, these relationships have been reconsidered and adjusted by the organisations. This raises questions about the uncertainty of the role of IC and its effectiveness within organisations, as well as doubts about the existence of POS among employees. COVID-19 undoubtedly changed and adjusted many aspects in everyone's life, including in the organisational world, highlighting the importance of studying issues such as: Has IC prevailed in organisation–individual relationships? Does IC trigger POS and IP? Does the existence of POS contribute to employee IP? Are IC, POS and IP correlated with each other? These questions are still recent, according to scholars, due to the limited information available regarding IC as a possible preserver of the quality of the organisation–individual relationship in the context of a pandemic. For these reasons, the present study is aimed at understanding how IC has been developed by organisations and how it has prevailed within them, taking into account the dramatic impact of COVID-19. Therefore, the following research hypotheses fall strongly on the relationships among IC, POS and IP:

**Hypothesis 1 (H1).** *Internal Communication, Perceived Organisational Support and Individual Performance are positively correlated with each other.*

**Hypothesis 2 (H2).** *Internal Communication is a predictor of Perceived Organisational Support and Individual Performance.*

In recent decades, IC has been a dimension that has contributed strongly to collaborator IP (Neto et al. 2018). According to Men and Stacks (2014), employees constitute a stronger force in organisations where their attitudes and behaviours contribute to productivity, IP, and organisational performance. However, due to the pandemic situation, the value of employees and their performance has been called into question. Organisations, in addition to having to focus on reinventing their business, should also be concerned about their employees with respect to POS and IP. Although many authors have tried to establish a clear definition of performance and its predictors and consequences, debate is still ongoing in the academic literature (Montes et al. 2003; Ribeiro et al. 2021). However, Pang and Lu (2018) proceeded with the idea that for an organisation, performance is one of the ways by which the extent of its effectiveness can be measured, as well as a vital aspect in the setting of goals and objectives. Swalhi et al. (2017) defined performance at work as the execution of the duties and responsibilities of a given function in which an individual's observable behaviours meet the conformities and objectives of the organisation.

Reflecting on the constituents of individual performance, Viswesvaran and Ones (2000) indicated that although there are different perspectives on what performance actually is, its evaluation, feedback and even merit pay systems are elements that help to understand IP. For this reason, IC is also suggested to be one of the main determinants of employee engagement, an association for which there are still only few empirical studies, despite its importance within the professional field (Karanges 2014). However, Osborne and Hammoud (2017) believe that engagement is important for both the employee and the organisation, is characterised by the involvement of all members of the organisation, and the appreciation of their behaviours as they express themselves physically, emotionally and cognitively during their function (Eldor and Vigoda-Gadot 2016). This organisational involvement awakens in employees a willingness to contribute to organisational success, which is consequently reflected emotionally through aspects such as well-being, positivity, and motivation (Osborne and Hammoud 2017), as well as satisfaction and productivity (Neto and Cruz 2017). In addition, support is also a factor that is positively correlated with IP, since there is empirical evidence showing that POS affects well-being and has a positive effect on satisfaction and CO, awakening a desire to remain with the organisation (Yücel et al. 2020).

It therefore makes sense to address the issue of productivity with respect to IP, which, although it is still an empirical issue (Vazire 2018), satisfactory and effective communication (Verčič and Špoljarić 2020) and the correct use of feedback (Song et al. 2017) are crucial elements in its understanding. However, with the evolution of technology and the easy access to it in recent years, organisations have opted for innovation and the use of technology as a tool to boost productivity and achieve institutional improvement (Meirinhos and Barreto 2018). According to Attaran et al. (2019), more technology-intensive workplaces are widely known for the optimisation of knowledge and boosting productivity. During the current pandemic situation, for which social isolation is the main prevention strategy, the use of technology has never made so much sense. Remote working has been employed by several organisations so that the normal functioning of their businesses could continue (Losekann and Mourão 2020). Furthermore, Venkatesh (2020) suggests that organisations, in the face of the pandemic, must develop effective strategies and organisational support as a form of help. The use of POS and IC is crucial in order to meet employees' emotional needs, such as respect and individual appreciation (Dursun 2015), and these are relevant factors that help to negotiate unprecedented times of crisis, such as that brought about by COVID-19 (Dirani et al. 2020). Following these considerations, it makes sense to analyse the following research hypothesis:

**Hypothesis 3 (H3).** *Perceived Organisational Support significantly mediates the relationship between Internal Communication and Individual Performance.*

### 3. Method

*Sample and Procedure*

Considering the hypotheses under study, a quantitative methodology was adopted, following a descriptive and interpretative approach of the data collected in a cross-sectional study. Regarding the data collection procedure, the questionnaire was developed using the Google Forms platform and made available on the Social Network Facebook, in order to allow the diverse admission of study participants. Since it was in the study's interest for data collection to occur during the lockdown, the questionnaire was accessible between 9 February and 15 March 2021.

For the construction of the instrument, the following indicators were considered in order to measure the variables under study, using a Likert agreement scale from 1 (totally disagree) to 5 (totally agree), in relation to the following statements:

Internal Communication: 6 items used, revealing a Cronbach's alpha of 0.92, based on Beaunoyer et al. (2020) and Lee et al. (2020). Example item: *"The organisation knew how to manage internal communication during lockdown"*;

Organisational Support: 8 items used, revealing a Cronbach's alpha of 0.91, based on Eisenberger et al. (1986). Example item: *"This organisation would ignore any complaint on my part"*;

Individual Performance: 4 items used, revealing a Cronbach's alpha of 0.90, based on Rego and Cunha (2008) and Staples et al. (1999). Example item: *"I am an effective employee"*.

The study sample was a convenience sample, being of an occasional non-probabilistic nature, and was composed of 340 participants. With respect to level of qualification, the sample included participants with various levels of qualification (38.2% secondary education/12th grade; 27.6% bachelor's degree; 12.4% master's degree), and with professions in diverse areas (46.2% Retail; 8.5% Health; 8.8% Manufacturing; 7.4% Education; 2.4% Sport; 2.4% Catering; 2.4% Agriculture; 1.8% Tourism; 22.4% Other area). There were more female respondents to the questionnaire, accounting for 67.6% of the sample, as opposed to males, who had a percentage of only 32.4%. With respect to age, the highest percentage fell in the age groups 36 to 40 years (20%) and up to 25 years (20%), followed by 31 to 35 years (19.7%), 26 to 30 years (15.9%), 41 to 45 years (12.4%), 46 to 50 years (6.5%), and 51 to 55 years (3.8%). With regard to educational qualifications, the highest percentage of respondents presented completed secondary education (38.2%), followed by bachelor's degree (27.6), master's (12.4%), graduate (7.9%), 9th year (7.6%), and 10th or 11th grade (3.8%). Regarding seniority in the company, it was observed that between 2 and 3 years (19.7%) and between 11 and 15 years (17.6%) were the answers with the highest percentage of respondents. However, these values were closely followed by 7 months to 1 year (13.8%), from 6 to 10 years (13.2%), from 4 to 5 years (11.1%), and from 16 to 20 years (9.4%). The options less than 6 months (7.3%) and more than 21 years (7.6%) were the least selected.

### 4. Results

An analysis of the data presented in Table 1 reveals that the studied variables are significantly associated with each other. In fact, it was found that IC is positively and significantly correlated with POS ($r = 0.48$; *sig.* $\leq 0.05$) and with the IP ($r = 0.24$; *sig.* $\leq 0.05$). Similarly, the variables POS and IP are also positively and significantly correlated ($r = 0.29$; *sig.* $\leq 0.05$). Thus, on the basis of this data, it can be concluded that there is a relationship between IC and POS, suggesting that Hypothesis 1 may be verified. Thus, the results support Hypothesis 1, indicating that the relationship between the variables under study are significantly and positively correlated with each other, as well as confirming Hypothesis 2, which predicted that the relationship between IC and IP would be significant and positive. These data confirm the presence of IC in the organisational environment, which,

in turn, generates support for the employee, which is ultimately positively reflected in their performance.

**Table 1.** Correlation matrix.

|  | Mean | St. Dev. | 1. | 2. | 3. | 4. | 5. | 6. | 7. |
|---|---|---|---|---|---|---|---|---|---|
| 1. AGE | - | - | 1 | - | - | - | - | - | - |
| 2.QUALIFICATIONS | - | - | −0.308 ** | 1 | - | - | - | - | - |
| 3. SENIORITY | - | - | 0.663 ** | −0.254 ** | 1 | - | - | - | - |
| 4.remote working | - | - | −0.026 | −0.303 ** | −0.011 | 1 | - | - | - |
| 5. SUPPORT | 3.3089 | 0.87936 | −0.038 | 0.001 | −0.129 * | −0.039 | 1 | - | - |
| 6. PERFORMANCE | 5.5158 | 1.02268 | −0.003 | −0.024 | 0.014 | −0.096 | 0.286 ** | 1 | - |
| 7. IC | 3.4529 | 0.95509 | 0.043 | 0.055 | −0.011 | −0.150 ** | 0.480 ** | 0.236 ** | 1 |

**. The correlation is significant at level 0.01 (2 extremities). *. The correlation is significant at level 0.05 (2 extremities).

To investigate the mediation effect described in Hypothesis 3 of the present study, which predicted the existence of a mediation effect of POS on the relationship between IC and IP, the methodological procedure proposed by Baron and Kenny (1986) was used. According to the authors, in order to test for a mediation effect, the following regression equations need to be performed: (1) regression of the mediator variable with the predictor variable; (2) regression of the criterion variable on the predictor variable; (3) regression of the criterion variable on the predictor variable, controlling the mediator variable. Thus, to verify the existence of a mediation effect, the following conditions must be met: (1) the predictor variable must affect the mediating variable in the first regression equation; (2) the predictor variable must be affected by the criterion variable in the second equation; (3) the mediating variable must affect the criterion variable in the third equation. Thus, when verifying these conditions, we may be presented with a partial mediation or a total mediation, depending on whether the effect of the predictor variable on the criterion variable remains significant or not when the mediating variable is present in the equation. Partial mediation is verified when the effect of the predictor variable on the criterion variable in the third equation is smaller than that in the second equation, whereas total mediation is verified when the effect of the predictor variable on the criterion variable in the third equation is no longer meaningful (Baron and Kenny 1986).

When applying the methodology in the present study, the first equation evaluated the effect of IC on PCOS ($\beta = 0.480$; *sig.* $\leq 0.01$), indicating a significant and positive relationship. The second equation verified the effect of IC on IP ($\beta = 0.231$; *sig.* $\leq 0.01$), and the third equation added the mediating variable, POS, into the analysis model ($\beta = 0.231$; *sig.* $\leq 0.01$). It was verified that when the mediating variable was predicted in the estimated regression, the relationship between Internal Communication and Individual Performance was no longer significant ($\beta = 0.231$; *sig.* $\leq 0.01$/$\beta = 0.121$; *sig.* $\geq 0.05$). This verifies the assumption of the presence of a total mediation effect of POS on the relationship between IC and IP. The second and third steps of this analysis are systematically presented in the Table 2, with the inclusion of control variables that seemed relevant to integrate into the scope of the study, such as age, gender, qualifications, and the modality of the respondent in a remote working situation.

**Table 2.** Regression Table.

| | MOOdelo | B | Error | Beta | t | *Sig.* |
|---|---|---|---|---|---|---|
| | (Constant) | 5.816 | 0.488 | – | 11.908 | 0.000 |
| | Gender | 0.242 | 0.127 | 0.108 | 1.900 | 0.058 |
| 1 | Age | −0.019 | 0.034 | −0.032 | −0.547 | 0.585 |
| | Educational qualifications | −0.030 | 0.047 | −0.039 | −0.630 | 0.529 |
| | Remote working | −0.243 | 0.147 | −0.099 | −1.651 | 0.100 |
| | (Constant) | 4.829 | 0.532 | | 9.081 | 0.000 |
| | Gender | 0.246 | 0.124 | 0.110 | 1.988 | 0.048 |
| 2 | Age | −0.024 | 0.033 | −0.042 | −0.734 | 0.463 |
| | Educational qualifications | −0.034 | 0.046 | −0.046 | −0.749 | 0.454 |
| | Remote working | −0.159 | 0.145 | −0.065 | −1.100 | 0.272 |
| | IC | 0.254 | 0.061 | 0.231 | 4.163 | 0.000 |
| | (Constant) | 4.348 | 0.536 | | 8.111 | 0.000 |
| | Gender | 0.237 | 0.121 | 0.106 | 1.954 | 0.052 |
| | Age | −0.017 | 0.033 | −0.029 | −0.511 | 0.609 |
| 3 | Educational qualifications | −0.032 | 0.045 | −0.043 | −0.715 | 0.475 |
| | Remote working | −0.169 | 0.142 | −0.069 | −1.193 | 0.234 |
| | IC | 0.132 | 0.068 | 0.121 | 1.950 | 0.052 |
| | Support | 0.272 | 0.072 | 0.231 | 3.773 | 0.000 |

Dependent variable: performance.

Thus, analysis of this data supported the hypotheses proposed in this study. On the basis of the results presented, it can be concluded that IC is associated with POS, meaning that organisations that invest in IC stimulate feelings of support and well-being, since this relationship was demonstrated to be significant and positive. Even within the pandemic context of COVID-19, the quality of organisation–individual relationships has been preserved through the implementation of IC strategies and with the help of POS. The results also reveal that IC is associated with IP, so regarding employees on the part of organisations was present in such a way that it was reflected in employee IP. Subsequently, it can also be concluded that the support transmitted to respondents was well associated with performance and productivity in the workplace in a pandemic context, including in the case of employees who were forced to work from home. Although the emergence of COVID-19 has disrupted several businesses, these results show that the investment in IC in the context of this pandemic has proved to be compensatory both in terms of its positive and significant effects on support, and in terms of performance. IC, in addition to the direct effects verified with respect to IP, was also able to activate a more distal effect, triggering a process that revealed the development and perception of support provided by the organisation to its employees, resulting in an impact on the productivity of employees.

It can thus be seen that, although COVID-19 has affected both the personal and professional lives of employees, organisations have been able to use IC as a form of communication and support, to the extent that they engaged with and helped their employees in such a way that this support was reflected in their performance, productivity, and well-being.

## 5. Conclusions, Limitations and Future Studies

Towards the end of 2020, an unprecedented pandemic, COVID-19, swept through communities around the world, leading to forceful organisational adaptation. Highly contagious, the disease has caused hundreds of thousands of deaths, tested the limits of health systems, and forced several periods of quarantine as a way of halting its spread. In this pandemic scenario, in which caution is necessary surrounding human contact, organisations have had to develop adaptive measures to guarantee the continuation of their business. Thus, organisations are faced with a long road ahead, where, in addition to concerns regarding the redefinition of organisational strategies, they are faced with crucial need to support employees in terms of both well-being and performance. In this context,

the relevance of the analysed indicators—IC, POS and IP—to the preservation of the quality of the organisation–individual relationship in a pandemic context is highlighted.

The evidence from the data indicates that IC is associated with POS and IP, which means that organisations that invest in IC stimulate a feeling of support and well-being, which is reflected in IP. This relationship supports the idea put forward by Neto et al. (2018) and Xiao (2020) that when an organisation engages with employees through fair rewards and POS, whereby an IC strategy is used and worked on in this process, employees tend to exhibit improved IP and become more dedicated to their work and the organisation.

In view of this conclusion, the existence of an association between these three factors and organisational strategy makes them essential to achieving the goals and objectives imposed by the latter. Furthermore, they contribute to employees' emotional well-being, allowing individuals to consider themselves part of the organisational family. IC works as a central process that allows employees to share information, and create ties and relationships (Neto and Cruz 2017), which consequently helps them to identify with and embrace the organisation's culture, values, beliefs and goals (Yue et al. 2020). The use of IC as a strategic form within the organisation fosters involvement with and concern for the desired goals of the organisation (Brandão 2018). However, in order for the goals to be achieved, it is necessary for POS to be reflected in employees' IP. Although conclusions are sparse in this area, this study concludes that while the COVID-19 situation has affected many lives and businesses, IC has been shown to positively compensate or this through its positive and significant effects on support through POS and on IP. For Yücel et al. (2020), POS contributed significantly to an increase in IP, as it is able to explain attitudes and behaviours, with the value of this support being reflected in employees' engagement and well-being. As a result, the results obtained demonstrate that IC is able to activate a more distal effect, triggering processes that reveal the development and perception of support provided by organisations to their employees and, as a result, suggesting that these have an impact on their productivity.

In view of the conclusions supporting our hypotheses, the data allows us to state that organisations have managed to employ and profit from IC in order to preserve the quality of organisation–employee relationships, even in a pandemic context. The organisational strategies used resulted in feelings of security and support among the respondents, indicating the presence of POS and, as a factor reflecting the effective work performed by the organisations, employees maintained good IP at work, even when working remotely. In short, organisations were able to adapt to the new reality and include employees in their strategy, providing them with the support and comfort they needed to deal with the disastrous consequences of COVID-19.

Within this framework, and taking into account the results obtained and conclusions drawn from them, the present study can be regarded as an important contribution to improving the existing theoretical knowledge underlying the vast area of IC, which remains a largely empirical issue among scholars. Furthermore, the study situates IC and the organisation–individual relationship within the context of COVID-19 in a positive way; in this sense, the present empirical study may be an asset in areas of this nature, due to the lack of existing analysis and the novelty of the disease. In this sense, for managers, this topic area could form part of practical learning with the aim of improving the relationship between their organisation and its employees, thus combatting the dramatic impact of COVID-19 on the communication, support, performance, well-being and satisfaction of all employees belonging to the organisation.

Although the present study draws a conclusion presenting positive results, this interpretation cannot be generalised, as the sample used was collected through social networks, and the shared survey reached a number of respondents from different regions and areas, who were therefore not subject to linear supervision. The data collection procedure was developed in consideration of the fact that the use of a non-probabilistic sample has limitations with respect to generalisability that need to be noted when reading the results of the study. Notwithstanding the data and procedures used in data collection and analysis, this

study was not totally immune to error. Future research should prospectively be developed using dyads as a way of ensuring a robust sample when dealing with these research issues.

From a theoretical perspective, although the study was guided and justified by existing information, the analysis of the correlations among the presented research components limited, since there has been insufficient empirical study. Regarding the collection of IC information, evaluation was possible on the basis of some scientific articles. However, with respect to IC and the impact of COVID-19, there was a critical lack of information, since we are dealing with a new and delicate subject in society. For this reason, linking IC with COVID-19 was a major limitation and, for this reason, a major challenge to the present empirical study, as this was the central theme of the study.

Regarding the use of this work for future studies, it would be pertinent to approach this theme in the context of specific audiences and/or organisations, as well as in different markets, in order to evaluate and generalise the results obtained here. From another point of view, and as a follow-up to this work, it would be interesting to study and evaluate the organisations that have successfully adapted to COVID-19 and the ways in which they strategically used IC and the results obtained within the environment in question. Future studies should also try to establish a confirmatory-oriented approach, which would add significantly to this exploratory study. Accordingly, we would like to suggest that future researchers develop models of analysis on a confirmatory basis regarding the relation between IC and relevant individual–organisation indicators, such as commitment, justice, trust and leadership. Confirmatory models for these issues should specifically be seen as the next step in research, as a way of further developing this research area.

**Author Contributions:** Conceptualization, D.R.G., P.L. and N.R.; methodology, D.R.G. and P.L.; software, D.R.G. and P.L.; validation, D.R.G. and N.R.; formal analysis, D.R.G. and N.R.; investigation, D.R.G., P.L. and N.R.; resources, D.R.G. and N.R.; data curation, D.R.G. and P.L.; writing—original draft preparation, P.L.; writing—review and editing, D.R.G. and N.R.; visualization, D.R.G. and N.R.; supervision, D.R.G.; project administration, D.R.G. and N.R. All authors have read and agreed to the published version of the manuscript.

**Funding:** This research was funded by FCT—Portuguese Foundation for Science and Technology—UIDB/04928/2020.

**Institutional Review Board Statement:** The study was conducted according to the guidelines of the Declaration of Helsinki.

**Informed Consent Statement:** Informed consent was obtained from all participants involved in the study.

**Data Availability Statement:** The data that support the findings of this study are available on request from the corresponding author, Daniel Roque Gomes, drmgomes@esec.pt.

**Conflicts of Interest:** The authors declare no conflict of interest.

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
