# Peer review of "When COVID-19 Is the Invader and Internal Communication Is the Hero: Understanding the Influence of Internal Communication on Individual Performance and Evaluating the Mediating Role of Perceived Support"

_admsci, doi:10.3390/admsci11040136_

Round 1

Reviewer 1 Report

Currently, the situation according to Covid-19 is new for all society, for entrepreneurs, politicians, people. Everybody must deal with it. It is very important to know, how pandemic situation affect the companies and the communication flows.

This study evaluate the impact of communication  in companies on the quality of the individual-organization relationship during the pandemic situation.

The research are is very actual.  The whole article is interestingly processed. I highly appreciate the structure of the article. It is clear, the individual parts follow each other.

The methodology, hypothesis are clearly stated. The set methods take into account the researched issues. For the research in the specified area, the distribution of the researched factors was correct. At the end of the research, the authors reached the given conclusions. The list of resources is appropriate to the research area and, of course, very actual.

I want the authors to get into the article, why the Facebook network was chosen for this research? What are the advantages of this network, according to research topic? Are you not afraid to reduce the credibility of the research by posting it on Facebook? Please, justify your choice.

Author Response

We would like to thank the reviewer for the comments regarding the study. We are very glad that the paper has received such favorable observations by the reviewer. Thanks in advance!

Regarding the question raised by the reviewer, on

“I want the authors to get into the article, why the Facebook network was chosen for this research? What are the advantages of this network, according to research topic? Are you not afraid to reduce the credibility of the research by posting it on Facebook? Please, justify your choice”

The study is an exploratory one, based on a non-probabilistic sample. The issue of having the data collection procedure based on Facebook relates to the high amount of social network user we can find here. We had no concerns on using social networks regarding credibility. However, we find useful to clarify the limitations of using social networks regarding data collection procedure.

Based on the question raised, we have inserted the following addition in the limitations section:

Although the present study has a conclusion of positive results, its reading should not be interpreted in a general way, as the sample used was collected through social networks, and the shared survey reached several respondents, from various regions and areas, therefore not being subject to linear supervision. The data collection procedure was developed with the consideration that using a non-probabilistic sample has generalization limitations that should be noted while reading the results of the study. Notwithstanding the data and procedures used in data collection and analysis, this study is not totally immune to errors.

Reviewer 2 Report

Thank you for offering me the opportunity to review this exciting study.

I found the topic very interesting, even if there are limits in the construction of the sample that can affect the validity of the results. These limitations should at least be highlighted to facilitate future research developments.

For example, the authors do not take into account the respondents' nationality. This aspect can be a significant aspect given that the governments of the various countries, and consequently the companies, have adopted different measures to contain the pandemic. Furthermore, the authors do not specify whether the sample of 340 respondents was pre-established or is the result of a generic request sent via Facebook, as defined by the authors, in an undifferentiated way. In this case, this could be another limitation that the authors should highlight in discussions and conclusions.

Furthermore, I suggest that the authors express the research question clearly in the introduction. It was not easy for me to understand the objective or objectives of the research.

Finally, a last critical aspect noted concerns paragraph 3. The authors do not specify the items used for the chosen indicators. Understanding the validity of the individual elements analyzed and the references to the literature based on which the authors identified them must be made explicit to give credibility to the research and the results.

Good luck with your work.

Author Response

We would like to thank the reviewer for the comments regarding the study. Thanks in advance!

Regarding the following comment: “I found the topic very interesting, even if there are limits in the construction of the sample that can affect the validity of the results. These limitations should at least be highlighted to facilitate future research developments.” and  “Furthermore, the authors do not specify whether the sample of 340 respondents was pre-established or is the result of a generic request sent via Facebook, as defined by the authors, in an undifferentiated way. In this case, this could be another limitation that the authors should highlight in discussions and conclusions”.

We have inserted the following sentence in the limitations section:

Although the present study has a conclusion of positive results, its reading should not be interpreted in a general way, as the sample used was collected through social networks, and the shared survey reached several respondents, from various regions and areas, therefore not being subject to linear supervision. The data collection procedure was developed with the consideration that using a non-probabilistic sample has generalization limitations that should be noted while reading the results of the study. Notwithstanding the data and procedures used in data collection and analysis, this study is not totally immune to errors. Future research developments should prospectively be developed using dyads as a way of having a robustness sample when dealing with these research issues.

Regarding the comment:

“Furthermore, I suggest that the authors express the research question clearly in the introduction. It was not easy for me to understand the objective or objectives of the research.”

We have added in the introduction the research question, as follows:

This study’s main research question is to understand the predictive effect of IC on IP, and whether POS has a mediating effect over this relationship within the pandemic context brought by COVID-19.

Regarding the comment:

Finally, a last critical aspect noted concerns paragraph 3. The authors do not specify the items used for the chosen indicators. Understanding the validity of the individual elements analyzed and the references to the literature based on which the authors identified them must be made explicit to give credibility to the research and the results.

We have identified on pg. 7 all the items origin with identification of the authors used, a sample item per variable, and also the cronbach alphas value for internal consistency purposes. All Cronbach Alphas show solid values, ranging from .90 to .92.

Reviewer 3 Report

The paper is very interesting and quite significant for the advancement of the management thematic field.

The issues of internal communication, individual performance and perceived organizational support are key issues that need to be addressed.

The "weakness" of the paper lies on the the statistical analysis. Even though the paper deploys authors' thoughts quite interestigly, tha analysis does not provide the fundametal basis to discuss findings. The hypotheses and model are not quite satisfactory explained. I would strongly suggest to improve the statistical analysis using Structural Equation Analysis and specifical focus on path analysis, in order to delve further and discuss findings.

I give merit to the authors and I will review in favor of the paper, but I would strongly suggest to work more on the statistical analysis and research methodology.

Author Response

We would like to start by addressing a thank you note the reviewer for the comments regarding the study.

Regarding the comment:

The "weakness" of the paper lies on the the statistical analysis. Even though the paper deploys authors' thoughts quite interestigly, tha analysis does not provide the fundametal basis to discuss findings. The hypotheses and model are not quite satisfactory explained. I would strongly suggest to improve the statistical analysis using Structural Equation Analysis and specifical focus on path analysis, in order to delve further and discuss findings.

We understand the comment and the idea surrounding the comment. We understand the value that a confirmatory study using SEM could bring to the study. However, this study in particular was developed under a different premise. This purpose of the study is purely exploratory. The study delivers a novice idea regarding Internal Communication during COVID times, which is a new issue of research on organizational studies. We have no record of several identical studies regarding the Internal Communication-Individual Performance that could be used as a solid base to ground a confirmatory study with the safety of researching on a non-exploratory orientation. This was our main guidance for the study.

However, we understand that the recommendation is significant, and have inserted the following paragraph in the limitations section:

Future studies should also try to establish a confirmatory-oriented approach to add significantly to this exploratory study. In accordance, we would like to suggest for future researchers to develop models of analysis with confirmatory basis regarding the IC relation with relevant individual-organization indicators, such as commitment, justice, trust or leadership. Confirmatory models over these issues in specific should be seen as the next step of research as a way of conquering stronger grounds of research.

Round 2

Reviewer 2 Report

I am pleased to have helped improve the quality of the paper. I noticed that the suggestions proposed were accepted by the authors, who clarified the unclear points of the work and highlighted the limits and future steps of the research.